# Quantitative Analysis of the Human Semen Phosphorometabolome by ^31^P-NMR

**DOI:** 10.3390/ijms25031682

**Published:** 2024-01-30

**Authors:** Rebeca Serrano, David Martin-Hidalgo, Jon Bilbao, Ganeko Bernardo-Seisdedos, Oscar Millet, Luis J. Garcia-Marin, Maria Julia Bragado

**Affiliations:** 1Research Group of Intracellular Signaling and Technology of Reproduction (SINTREP), Research Institute INBIO G+C, University of Extremadura, 10003 Caceres, Spain; rebecasp@unex.es (R.S.); davidmh@unex.es (D.M.-H.); 2Precision Medicine and Metabolism Laboratory, CIC bioGUNE, Basque Research and Technology Alliance (BRTA), 48160 Derio, Spain; jbilbao@cicbiogune.es (J.B.); ganeko.bernardo@deusto.es (G.B.-S.); omillet@cicbiogune.es (O.M.); 3Department of Medicine, Faculty of Health Sciences, University of Deusto, 48007 Bilbao, Spain; 4CIBERehd, Instituto de Salud Carlos III, 28220 Madrid, Spain

**Keywords:** NMR-based phosphoromics, metabolomics, human sperm, seminal plasma, asthenozoospermia

## Abstract

Phosphorus-containing metabolites occupy a prominent position in cell pathways. The phosphorometabolomic approach in human sperm samples will deliver valuable information as new male fertility biomarkers could emerge. This study analyzed, by ^31^P-NMR, seminal plasma and whole semen from asthenozoospermic and normozoospermic samples (71% vs. 27% and 45% vs. 17%, total and progressive sperm motility, respectively), and also ejaculates from healthy donors. At least 16 phosphorus-containing metabolites involved in central energy metabolism and phospholipid, nucleotide, and nicotinamide metabolic pathways were assigned and different abundances between the samples with distinct sperm quality was detected. Specifically, higher levels of phosphocholine, glucose-1-phosphate, and to a lesser degree, acetyl phosphate were found in the asthenozoospermic seminal plasma. Notably, the phosphorometabolites implicated in lipid metabolism were highlighted in the seminal plasma, while those associated with carbohydrate metabolism were more abundant in the spermatozoa. Higher levels of phosphocholine, glucose-1-phosphate, and acetyl phosphate in the seminal plasma with poor quality suggest their crucial role in supporting sperm motility through energy metabolic pathways. In the seminal plasma, phosphorometabolites related to lipid metabolism were prominent; however, spermatozoa metabolism is more dependent on carbohydrate-related energy pathways. Understanding the presence and function of sperm phosphorylated metabolites will enhance our knowledge of the metabolic profile of healthy human sperm, improving assessment and differential diagnosis.

## 1. Introduction

The spermatozoon is a highly specialized cell with exceptional DNA compaction and is almost devoid of cytoplasm, in which there appears to be an absence of protein translation. In this context, where their transcriptional and translational activities are almost silenced, metabolites present in spermatozoa are particularly important in providing a direct snapshot of the male gamete’s cellular activity or physiological state. Mass spectrometry (MS) and nuclear magnetic resonance (NMR) spectroscopy are the analytical and complementary tools of choice in metabolomics research today [1,2,3]. These technologies applied in the field of male reproduction in humans are used to explore the patterns of metabolites in seminal plasma [4,5,6,7,8,9,10,11,12,13] and spermatozoa [14,15,16] to identify possible associated biomarkers to different conditions related to male infertility. Furthermore, the interest in understanding the sperm physiology underlying semen quality has led to the analysis of the endogenous metabolome in healthy donor samples as well [17,18,19,20]. In addition, recent investigations have shown that a metabolomic approach is helpful for the advancement of precision medicine, since it may be used as a prognosis tool to accurately predict the efficacy of a potential treatment (e.g., varicocelectomy or microTESE) for several male infertility causes [21,22]. 

The Human Metabolome Database 2022 (HMDB 5.0) has a collection of 217,920 annotated metabolites, interconnected and continuously processed by more than 18,000 enzymatic reactions integrated into some 26,000 metabolic pathways [23]. Moreover, metabolomics is commonly used to analyze all metabolites in biological systems simultaneously. However, complementary techniques are applied to reduce complexity and maintain maximum functional information [24]. For example, the combination of NMR with numerous stable isotopes, such as ^13^C-carbon or ^31^P-phosphorus, makes it possible to label the nutrients of interest, monitor pathway fluxes, and highlight specific metabolic processes [16,19,25,26]. Along these lines, recently, Reynolds and co-workers, aiming to study the relationship between energy metabolism and sperm motility in humans, investigated, by ^13^C-NMR, how spermatozoa from normozoospermic and asthenozoospermic ejaculates metabolized ^13^C-labeled substrates [16]. Instead, ^31^P-phosphorus is a naturally abundant and NMR-active isotope that does not require an external supplemental source. Moreover, phosphorus-containing metabolites are critical and key regulators of important cellular processes such as the energy metabolome, redox state, or cell signaling, and they may comprise 36% of the human metabolome [24]. Hence, the study of alterations in the phosphorylated fraction of the cell metabolism, the “phosphorome” [27], which can point out molecules containing only phosphorus atoms in the complex mixture of metabolites, is compelling as an indicator of an imbalance in the normal functioning of the cell. In mammalian male reproduction, previous studies have already applied the ^31^P-NMR technique to analyze the lipid composition or the energy metabolism of human, ram, goat, and bull spermatozoa [28,29], which has contributed to confirming the potential utility of the phosphorylated metabolome in the analysis of mammalian sperm physiology. 

Our previous study investigated the phosphoproteins underlying human sperm motility in two spermatozoa subpopulations with different motility using phosphoproteomics techniques [30]. Here, we extend the analysis and go further with a phosphorometabolomics approach to deepen our knowledge of the possible differential phosphorus-containing metabolites linked to low sperm motility using a comparative ^31^P-NMR analysis of human seminal plasma samples from asthenozoospermic and normozoospermic samples. As mentioned above, phosphorus-containing metabolites are critical modulators of important cellular processes; therefore, we have examined other sperm functional parameters such as viability, mitochondrial membrane potential, and mitochondrial superoxide production. In addition, we also identified the most relatively abundant phosphorus-containing metabolites in seminal plasma and spermatozoa from ejaculates from healthy human donors. The results of this work will contribute to establishing the phosphorometabolomic landscape in human semen samples and defining alterations in the phosphorus-containing metabolites pattern. This challenge will increase the understanding of the mechanisms that regulate cellular biochemistry in the human spermatozoon and, therefore, the possible underlying metabolic causes of male infertility. As far as we know, this is the first work that uses phosphorometabolomics to study the human sperm phosphorometabolome in normozoospermic and asthenozoospermic subjects.

## 2. Results

### 2.1. Sperm Quality Measures

Table 1 summarizes the sperm concentration and motility analyzed by CASA in both the NORMO and AST donors according to WHO [31]. The average sperm count was 47 million/mL in the AST group and 167 million/mL in the NORMO group, although the mean volume was 3 mL in both cases. The sperm motility parameters in the AST group showed that the average total motility was 27%, the progressive motility was 17%, and the rapid progressive spermatozoa was 11%, whereas, as expected, these percentages were significantly higher (71%, 45%, and 28%, respectively) in the NORMO group. To characterize the sperm functional differences between both groups, we analyzed some sperm parameters by flow cytometry. The scores in the AST group display a significantly lower sperm viability (46% AST vs. 70% NORMO). In contrast, there were no differences in the high mitochondrial membrane potential (49% NORMO vs. 44% AST) or mitochondrial superoxide anion production (31% NORMO vs. 37% AST) in our experimental conditions (Table 1).

### 2.2. Phosphoromic Analysis of Semen: General View

Our methodology to study phosphorus-containing metabolites is based on the direct observation and quantification of the phosphorus signals (^31^P-NMR) in ejaculates, seminal plasma (SP), and spermatozoa from humans. A representative comparison of the ^31^P-spectrum from regions 5 to −3 ppm obtained from the SP, shown in Figure 1, illustrates the peak dispersion and metabolite assignation. A list of all the detectable metabolites in the spectra can be found in Appendix A. Subsequently, quantitative data analysis was possible because the intensity of the signal directly correlates with the analyte concentration. Normalization to quantitatively compare the amounts of metabolites between the different samples was performed in the SP and whole semen per volume and is expressed as nmol/μL. For the spermatozoa analysis, the normalization of the abundance of each metabolite was performed using millions of cells and is defined as nmol/millions of cells. In Appendix A can be found a complete list of the information showing the metabolite concentration in each sample.

Our study assigned 16 phosphorus-containing metabolites covering almost all the central energy metabolism pathways (glycolysis and pentose phosphate pathway), glycogenesis, phospholipids, nucleotide, and nicotinamide metabolism pathways (Appendix A). However, a few peaks in the ^31^P-NMR spectra could not be assigned to any common metabolite and they were referred to as phosphodiester unknown 1–4 (PDE1–4). Additionally, one peak was assigned to more than one phosphorus-containing metabolite, sedoheptulose 7-phosphate (S7P) and fructose 1,6-bisphosphate (FBP). In these cases, they were left unsolved due to overlap and analyzed together as S7P and FBP, except when the phosphorometabolites were grouped according to the metabolic pathways in which they are involved.

The concentration range in which we detected the phosphorus-containing metabolites was in millimolar, with 9.85 nmol/μL being the highest mean concentration found, corresponding with glycerophosphocholine (GPC) and 0.01 nmol/μL being the lowest one, corresponding with PDE3. Through the ^1^H-NMR, six non-phosphorus-containing metabolites previously identified in human semen (choline, CHO; citrate, CIT; tyrosine, Tyr; lactate, LAC; glutamine, Gln and phenylalanine, Phe) were analyzed and a higher concentration range was quantified (from 25.05 to 1.53 nmol/μL, matching with CHO and Phe, respectively) (Appendix A).

To identify correlations between the phosphometabolites and seminal parameters in different conditions related to fertility, the bioinformatic tool MBROLE 2.0 [32] and the KEGG pathways database were consulted to group the phosphorometabolites according to the metabolic pathways in which they are involved. Appendix A shows the result from this classification where those involved in carbohydrate metabolism (Pentose phosphate pathway, Starch and sucrose metabolism, Glycolysis/Gluconeogenesis, and Pentose and glucuronate interconversions) and lipid metabolism (Glycerophospholipid and Inositol phosphate metabolism) stand out, which account for 30% and 41%, respectively, of the phosphorometabolites assigned.

### 2.3. Phosphoromics Analysis in Seminal Plasma from AST and NORMO Groups

To study the potential value of the phosphoromic approach, we investigated possible differences among the human seminal samples from different motility qualities as those obtained from the AST and NORMO groups. Figure 2 compares the mean value for each metabolite assigned in each group. Possible interindividual differences do not affect the data because no high deviations were observed for the analytes (Appendix A). Likewise, the average semen volume was 3 mL in both experimental groups (Table 1), and there was no correlation between the metabolite concentrations with the volume of the ejaculate in each sample (Appendix A).

The amounts of three phosphorometabolites, phosphocholine, glucose-1-phosphate, and, to a lesser degree, acetyl phosphate (PCh, G1P and AcP, respectively), were greater in the AST group (Figure 2a), showing significant differences for PCh (0.53 AST vs. 0.05 NORMO). In contrast, the abundance of three non-phosphorus-containing metabolites, CHO, Phe, and Tyr differed significantly between the AST and NORMO groups, being up to 1.5 times less abundant in the seminal samples from the asthenozoospermic men (Figure 2b). 

We also analyzed, by NMR, unprocessed ejaculates from each group. We observed almost 5 times more G1P and 1.5 times more AcP and PCh in the AST group compared with the NORMO group, although these differences were not statistically significant. However, CHO, Tyr, and Phe were statistically more abundant in the whole ejaculates from the NORMO group (Appendix A).

### 2.4. Metabolic Phosphorome Differences between Spermatozoa and Seminal Plasma from NORMO Group

To find the most representative phosphorometabolites in each human semen fraction, we compared the relative number of phosphorus-containing metabolites in the spermatozoa vs. SP from the NORMO group (Table 2). Considering the 20 phosphometabolites detected, 14 of them showed a greater relative abundance in the spermatozoa, whereas 6 were more abundant in the SP: glycerophospho(dimethyl)ethanolamine (GP(dimethyl)E), glycerophosphoinositol (GPI), nicotinamide adenine dinucleotide phosphate reduced form (NADPH), glucose-6-phosphate (G6P), 6-phosphogluconic acid (6PG), and GPC. Thus, the phosphorus-containing metabolite fraction is relatively higher in the spermatozoa (70% spermatozoa vs. 30% SP). In the spermatozoa fraction, the phosphorus-containing metabolites with the highest relative abundance were PDE3, S7P and FBP, xylulose-5-phosphate (X5P), Pch, G1P, and PD4 (Table 2), highlighting the importance of phosphorometabolites involved in carbohydrate metabolism related-pathways in human spermatozoa (Table 2). However, in the SP, the phosphorometabolites implicated in lipid metabolism, such as GP(dimethyl)E or GPI, exhibited a significantly higher relative abundance than in the sperm cells (Table 2). Appendix A shows the concentration of each metabolite in the spermatozoa samples. Appendix A shows the relative amounts of each metabolite in the spermatozoa and SP from each donor.

Regarding the relative quantities of the metabolites detected by ^1^H-NMR, the higher relative abundance of Gln, CIT, and LAC in the spermatozoa and CHO, Tyr, and Phe in the SP is noteworthy (Table 2). There were statistically significant differences for Gln, CIT, and CHO.

## 3. Discussion

This study is pioneering, as the phosphorometabolome of human semen from asthenozoospermic and normozoospermic donors by ^31^P-NMR has not been investigated. Cellular metabolism analysis is hard due to its complexity. However, examining only phosphorylated metabolites focuses research on specific molecules and the metabolic pathways interconnecting them. Phosphorylated metabolites are critical regulators of essential processes, such as energy metabolism, the redox state, or signaling. Furthermore, understanding male infertility pathophysiology is still challenging and identifying human semen phosphorometabolites as potential infertility biomarkers using the ^31^P-NMR technique will positively impact this field.

This phosphorome study of human semen by ^31^P-NMR identified 16 phosphorometabolites, of which 71% (X5P, 6PG, G6P, S7P, G1P, Pch, GPC, GPI, GPE, GP(N-biotin)E, GP (dimethyl)E, GP(monomethyl)E) are involved in carbohydrate and lipid metabolism-related pathways and 12% (NMP and NMPc) in signal transduction and purine metabolism. The ^31^P-NMR analysis detected 20 phosphorometabolites, but 4 (PDE 1–4) are unknown metabolites not included in spectral databases. Despite new advances, the HMDB contains up to 1,581,537 unannotated metabolites [33]. 

By comparing the abundance of the phosphorometabolites from the human seminal plasma between the normozoospermic and asthenozoospermic groups, AcP (slightly) and especially PCh and G1P were richer in lower-motility spermatozoa (AST). These differences are not likely due to the ejaculatory–abstinence period duration that is associated with the available amounts of metabolites present in seminal fluid [34], as in this study, it was always 3–4 days. Previously, higher PCh levels were also found in the asthenozoospermic seminal plasma [35]. Semen phosphatases can rapidly hydrolyze PCh to phosphate and choline [36]. Therefore, the higher PCh abundance and the lower CHO amount detected in the AST group may indicate PCh accumulation due to (i) abnormal choline metabolism, which is essential for the composition, fluidity, adhesion, and signaling of the sperm membrane, and it has already been related to asthenozoospermia [35] or (ii) altered phosphatase(s) activity in AST men.

Here, AcP abundance, a product of taurine and hypotaurine metabolism, is higher in the AST group. Taurine, one of the most abundant amino acids in the male reproductive system, plays several functions: antioxidant, anti-apoptotic, and osmotic and calcium regulation [37]. Although the AcP increase detected is very slight, this contrasts with the previous idea that low taurine levels in seminal plasma are a marker of male infertility [9] and with the recent data showing that taurine presence is related to higher human sperm motility [34].

We found a higher abundance of G1P in the seminal plasma of the AST group, which might be explained by phosphoglucomutase activity, already measured in human seminal plasma [38], after glucose phosphorylation by hexokinase I, which is contained in human prostasomes, prostate-release extracellular vesicles found in high concentrations in seminal plasma [39]. Further, increased amounts of glucose and inorganic phosphate were detected in the seminal plasma from humans with asthenozoospermia, teratozoospermia, and unexplained infertility [4,9,40]. Likewise, a proteomic-scale reconstruction of the human metabolic model proposes that the phosphoglucomutase gene may be implicated in asthenozoospermia [41]. In agreement, we found that G1P is 2-fold higher than G6P in seminal plasma in the AST group, which may indicate that phosphoglucomutase activity is higher in low-motility sperm, thus leading to a higher G1P amount. Another explanation is that G1P could be produced from the catabolism of glycogen in human semen, which provides about one sixth of the total glycolysable sugars [42]. In addition, the lower abundance (1.5-fold) of three non-phosphorylated metabolites (CHO, Phe, and Tyr) in the seminal plasma of the AST group agrees with previous works correlating them with poor sperm quality [7,10,35]. Therefore, we consider that altered levels of CHO, Phe, and Tyr together with PCh, G1P, and AcP in the seminal plasma could indicate that these metabolites play a fundamental role in human sperm motility and also that the metabolic pathways in which they participate (mainly those involving Phe, Tyr, CHO, and G1P) are clearly impaired in asthenozoospermic men.

Analogous results to seminal plasma were obtained when comparing the whole ejaculates, which is reasonable because spermatozoa only represent <10% of the total human semen volume [43], seminal plasma being the major component of the ejaculate. Thus, we conclude that the metabolism of spermatozoa in the whole human semen does not significantly contribute to the phosphometabolic profile that is observed in the seminal plasma.

Technically, ^31^P-NMR spectrometry and subsequent reliable metabolite quantification require at least 150 million human spermatozoa due to the relatively low NMR sensitivity [3,17,44]. The sperm concentration is generally poorer in asthenozoospermic ejaculates, which would have forced us to pool samples of identical phenotypes, leading to missing interindividual differences. Therefore, we first analyzed the phosphorus-containing metabolites in the normozoospermic ejaculates. The phosphorylated fraction of metabolites was higher in the spermatozoa (70% vs. 30% in seminal plasma) showing the highest relative sperm abundance of PDE3, S7P and FBP, X5P, Pch, G1P, NMP, and PD4, highlighting their relevance in energy pathways.

However, the phosphorometabolites implicated in lipid metabolism, such as GPC, GP(dimethyl)E, or GPI, were spotlighted in the seminal plasma. The highest GPC abundance in the seminal plasma confirms the ^31^P-NMR results from other mammalian semen, such as boar, ram, goat, and bull [28]. The seminal plasma contains molecules available to ejaculated spermatozoa and are responsible for maintaining proper motility [45]. Moreover, whereas sperm phosphoglyceride metabolism as an energy pathway is conceivable because phospholipases are present in seminal plasma coming from the secretions of the seminal vesicles, prostate, and epididymis [46,47,48], glycerophospholipids’ main role in the seminal plasma is probably related to the sperm membrane integrity, as GP(dimethyl)E and GPI are the principal glycerophospholipids of biological membranes. Therefore, their higher abundance in the human seminal plasma suggests that lipid transport or plasma membrane interaction might be essential for optimal sperm function. Thus, the available glycerophospholipids would be incorporated into the sperm membranes, modulating their fluidity whenever needed. Technically, the metabolite differences between the spermatozoa and seminal plasma indicate an appropriate sample preparation and insignificant mixing between them. However, we are aware that ejaculates might contain a negligible number of other cells that are metabolically active (leukocytes and epithelial cells) and that possible spermatozoa lysis might have occurred before spermatozoa isolation from the seminal plasma. Both situations could slightly affect our results. Moreover, future studies are needed considering our sample size and the fact that some phosphorometabolites in low concentrations were undetected and that ^31^P-NMR application is restricted for some phosphorus-containing metabolites (PDE 1–4). Thus, PDE 1–4 identification by MS could be an interesting advance in the human sperm phosphorome, especially PDE3 and PDE4, because they are more abundant in spermatozoa (by 12 and 2 times, respectively).

## 4. Materials and Methods

### 4.1. Human Semen Samples

Ejaculates from 13 donors obtained by masturbation into sterile containers after sexual abstinence (3–4 days) were used. Semen samples were selected according to the World Health Organization (WHO) recommendations from normozoospermic (NORMO) and asthenozoospermic (AST) [31]. After complete liquefaction (30 min^−1^ h), samples were processed and sperm volume, concentration, and motility (%) were evaluated by a computer-assisted semen analyzer (CASA) following WHO [31].

### 4.2. Human Spermatozoa Motility Evaluation

From each ejaculate, 6 μL was placed in a 37 °C pre-warmed Spermtrack sperm counting chamber (Proiser R+D, Paterna, Valencia, Spain). Spermatozoa images were taken using a microscope equipped with a 10X negative-phase contrast objective, a heated stage, and a CCD camera, evaluating at least 300 spermatozoa/sample. Digitalized images were analyzed using ISAS^®^ system (Integrated Semen Analysis System, Proiser R+D, Paterna, Valencia, Spain).

### 4.3. Analysis of Human Sperm Functional Parameters by Flow Cytometry

Semen samples were analyzed as previously described [30]. Briefly, 200,000 spermatozoa/ejaculate were incubated in Sperm Washing Medium (IrvineScientific, Daimler, St. Santa Ana, CA, USA) in darkness at room temperature (RT) with SYBR-14 (20 nM) and PI (9.6 μM) for 20 min to measure sperm viability. For measuring mitochondrial superoxide anion production, samples were incubated for 15 min with MitoSOX™ Red (2 μM, 37 °C, 5% CO_2_) and 10 min with MitoTracker™ Deep Red FM (50 nM, 37 °C) to evaluate mitochondrial membrane potential.

A NovoCyte TM flow cytometer (ACEA Biosciences, Inc., San Diego, CA, USA) and ACEA NovoExpress 1.2.1 TM software were used. All probes were from Thermo Fisher Scientific (Eugene, OR, USA). The fluorescence values of SYBR-14 were collected in the laser-excited fluorescence channel (BL1) using a 525 nm band-pass filter, whereas MitoSOX™ fluorescence was collected in the BL2 channel using a 585 nm band-pass filter and PI fluorescence and MitoTracker™ were collected in the BL3 channel using a 620 nm band-pass filter. The results were expressed as the average percentage of labelled spermatozoa for each parameter analyzed ± standard error of the mean (SEM). Sperm viability was considered the percentage of SYBR-14+ and PI-labeled cells. Superoxide anion production in the mitochondria of live sperm cells was examined as the percentage of MitoSOX™ and the population of spermatozoa with a high mitochondrial membrane potential was considered as the percentage of MitoTracker™.

### 4.4. Human Sperm Samples Preparation

Human ejaculates were divided into seminal plasma (SP) and cells (spermatozoa) by centrifugation (RT, 4 min, 900× *g*). Spermatozoa (pellet) and seminal plasma (supernatant) (100 μL) were recovered, centrifuged (RT, 1 min, 10,000× *g*), and left on liquid N2 before storage (−80 °C). CASA analysis showed that typically ≤1% of pellet cells were non-sperm cells. For whole ejaculate analysis, 100 μL from each donor was frozen on liquid N2 and stored (−80 °C).

### 4.5. NMR Sample Preparation

Samples were slowly defrosted for 5 min within the ice. A volume of 1.3 mL of CHCl_3_:MeOH:ddH_2_O in a ratio 41.7:35.6:32.7 (*v*/*v*/*v*) were added to each sample. For SP and unprocessed ejaculated samples, 100 μL of volume was added to the previous triphasic mixture. In all the cases, 254 nmol of sodium trimethylsilylpropanesulfonate (DSS) as internal extraction control was added. The 1.5 mL tubes with the previous mix were placed at 4 °C with agitation for 4 h. The mixture was centrifuged at 4 °C at 30,000× *g* for 30 min. The upper phase of the dissolution (hydrophilic phase) and the down phase of the dissolution (lipophilic phase) were transferred to new 2 mL tubes. All the samples were then dried overnight in a Speed-Vac. 

Preparation of samples to be subjected to the NMR includes mixing the extracted metabolites from the hydrophilic phase with 300 μL of a buffer solution (0.317 mM TMP+ (tetramethylphosphonium chloride), 1 mM EDTA, 0.5 mM Gadobutrol, 200 mM Tris-d6, all in D2O with a pH = 7). For the lipophilic phase, 300 μL of DMSO-d6 was added with TPP (triphenylphosphine oxide) at 4 mM. After vortex, 300 μL was added into a 5 mm Shigemi NMR tube. In all cases, sample preparation was manually performed at RT. TMP+ and TPP were added as ^31^P reference compounds.

### 4.6. NMR Measurements

All NMR experiments were recorded at 298 K on a Bruker 600 MHz (12 T) Avance III spectrometer equipped with a BBO (^1^H,^31^P) probe head. For each sample, three different experiments were collected: (i) 1D ^31^P-NMR zgig spectrum with inverse gated ^1^H decoupling; (ii) 1D ^1^H-NMR p3919gp with water signals suppression using a binomial 3-9-19 pulse with echo gradient pair; and (iii) 2D ^1^H-^31^P-HSQMBC-TOCSY spectrum with coherence selection by gradients. The 1D ^31^P experiment was recorded with long interscan delays d1 > 3∙T1, max(^31^P) for quantification, where the paramagnetic relaxation enhancer (0.5 mM of Gadobutrol) added only to the hydrophilic fraction allowed for the use of a short d1 = 1 s while the lipophilic fraction required d1 = 5 s.

### 4.7. Identification and Quantification of Metabolites

All NMR data processing and analysis were performed using TopSpin 4.0.7 (Bruker Biospin GmbH, Ettlingen, Germany) and in-house MatLab scripts. Assigned ^31^P-metabolites were quantified by referencing their ^31^P peak integral against the added internal reference compound. In signal overlap, we applied peak deconvolution (command LDCON) to assign the corresponding peak areas and thus determine the final concentration. Internal reference can easily be found on the left in the phosphorus spectra, which will be integrated and calibrated to a determined concentration value (0.317 mM for TMP+ and 4 mM for TPP). Considering this, metabolic quantification is performed by peak integration and comparison with the reference considering the spin system of the peak. In the lipophilic phase, no metabolites were observed due to their low concentration, below the detection limit.

For the ^1^H spectra, we identified, integrated, and calibrated the reference’s peaks. We performed the same operation for phosphorus metabolites but, in this case, for lactate (LAC), citrate (CIT), choline (CHO), tyrosine (Tyr), glutamine (Gln), and phenylalanine (Phe) quantification in the hydrophilic phase. These metabolites have already been identified by NMR spectroscopic approaches in human semen [9,10,35,49,50] and are used here to ensure a proper analytical procedure.

### 4.8. Statistical Analysis

The IBM Corp. software, IBM SPSS Statistics for Macintosh, Version 27.0 (Armonk, NY, USA), was used for statistical analysis. The mean, STD (standard deviation), and SEM were calculated for descriptive statistics. Data were tested for normal distribution with a Shapiro–Wilk test and homoscedasticity with a Levene’s test. Differences in sperm motility and functional parameters between AST and NORMO samples were determined by a parametric *t*-test (Student’s *t*-test). To identify significant differences between phosphorometabolites in the groups, a Student’s *t*-test or a Mann–Whitney U-test (when the data were not normally distributed) was performed. Statistical significance was accepted at *p* < 0.05.

## 5. Conclusions

The analysis of the human semen phosphorometabolome reveals that the phosphorometabolites produced by the metabolism of carbohydrates in human spermatozoa are more abundant than those produced by lipid metabolism. In fact, our results highlight specific metabolic alterations in spermatozoa with low motility, where an increase in the phosphometabolites acetyl phosphate, phosphocholine, and glucose-1-phosphate is found. This finding suggests potential impairments in choline, taurine, and glycerophospholipid metabolism occurring in the asthenozoospermia condition, a known cause of human male infertility.

Further research is needed on the metabolic pathways involving the mentioned phosphometabolites to discover which enzyme or reaction is impaired in asthenozoospermic men. In addition, more studies are necessary on the sperm phosphorometabolome that is involved in other sperm functional processes key to achieving fertilization, such as capacitation. Thus, a better understanding of the presence and function of phosphorylated metabolites in human sperm is expected to increase scientific knowledge about the metabolic profile of healthy human sperm, identifying potential molecular biomarkers of human sperm health, which in turn will upgrade the evaluation and differential diagnosis of infertile men and could ultimately simplify the selection of the best in vitro treatment to overcome infertility.

## Figures and Tables

**Figure 1 ijms-25-01682-f001:**
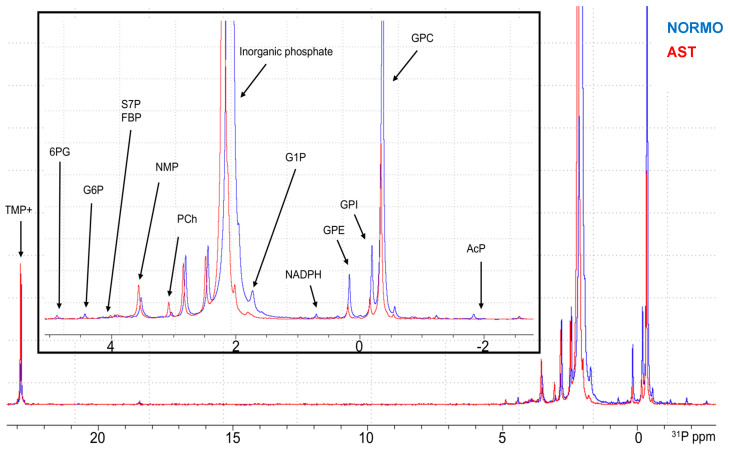
^31^P-NMR spectra representative of seminal plasma (SP) from asthenozoospermic (AST in red) and normozoospermic (NORMO in blue) groups. Regions 5 to −3 ppm with respective metabolites assignation are shown. Note: 6PG = 6-phosphogluconic acid; G6P = glucose 6-phosphate; S7P = sedoheptulose 7-phosphate; FBP = fructose 1,6-bisphosphate; NMP = nucleotide monophosphate; PCh = phosphocholine; G1P = glucose 1-phosphate; GPE = glycerophosphoethanolamine; GPI = glycerophosphoinositol; GPC = glycerophosphocholine; AcP = acetyl phosphate; TMP+ = tetramethyl phosphonium chloride (NMR reference compound).

**Figure 2 ijms-25-01682-f002:**
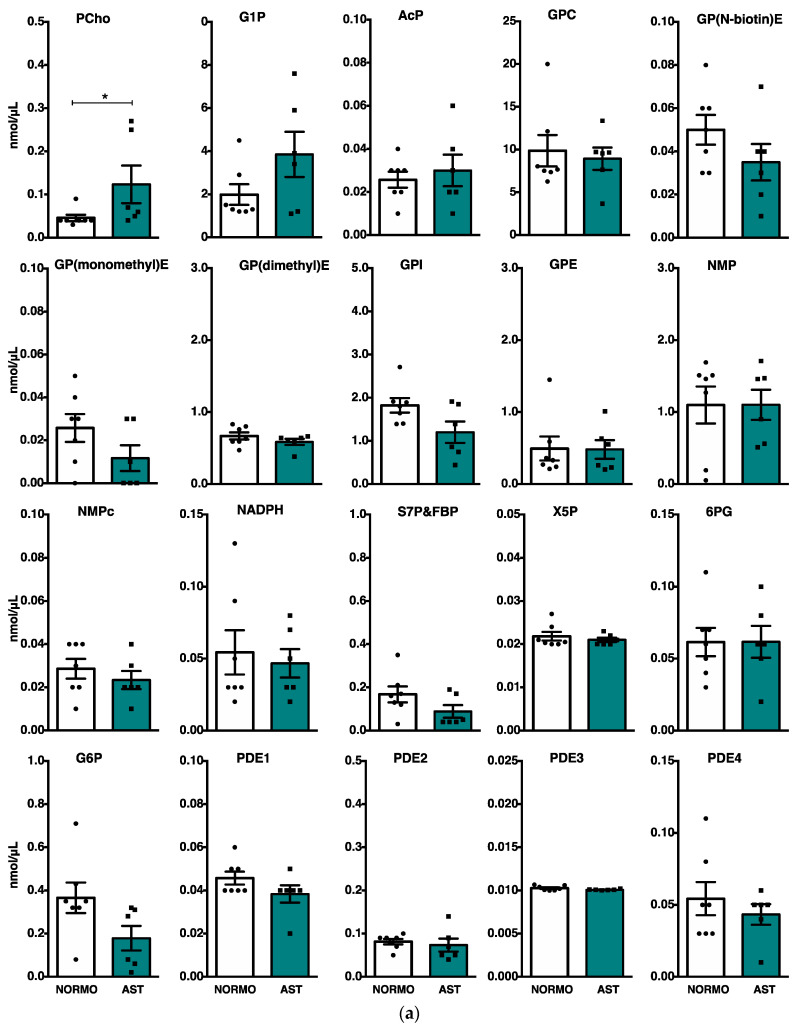
Comparison of quantified metabolites (nmol/μL) obtained from NMR spectrum in seminal plasma between NORMO (white bars; *n* = 7) and AST (in color bars; *n* = 6) groups; bars represent the average ± SEM. * *p* < 0.05. (**a**) Metabolites obtained from ^31^P-NMR spectrum; (**b**) six metabolites obtained from ^1^H-NMR spectrum. * *p* < 0.05. Note: PCho = phosphocholine; GP1P = glucose 1-phosphate; AcP = acetyl phosphate; GPC = glycerophosphocholine; GP(N-biotin)E = Glycerophospho(N-biotin)ethanolamine; GP(monomethyl)E = Glycerophospho(monomethyl)ethanolamine; GP(dimethyl)E = Glycerophospho(dimethyl)ethanolamine; GPI = glycerophosphoinositol; GPE = glycerophosphoethanolamine; NMP = nucleotide monophosphate; NMPc = cyclic nucleotide monophosphate; NADPH = reduced nicotinamide adenine dinucleotide phosphate; S7P = sedoheptulose 7-phosphate; FBP = fructose 1,6-bisphosphate; X5P = xylulose 5-phosphate; 6PG = 6-phosphogluconic acid; G6P = glucose 6-phosphate; PDE1–4 = phosphodiester unknown 1–4; CHO = choline; CIT = citrate; Tyr = tyrosine; LAC = lactate; Gln = glutamine; Phe = phenylalanine.

**Table 1 ijms-25-01682-t001:** Seminal parameters of the human donors of this study.

	NORMO	AST
Semen volume (mL)	3 ± 1	3 ± 0
Sperm concentration (10^6^ per mL)	167 ± 29	47 ± 11 **
Total sperm motility (%)	71 ± 2	27 ± 5 **
Progressive motility (%)	45 ± 4	17 ± 4 **
Rapid progressive (%)	28 ± 3	11 ± 3 **
Sperm viability (%)	70 ± 3	46 ± 2 **
hMMP (%)	49 ± 4	44 ± 9
SOP (%)	31 ± 7	37 ± 2

Note: NORMO = normozoospermic donors; AST = asthenozoospermic donors; hMMP = high mitochondrial membrane potential; SOP = mitochondrial superoxide anion production. Data, expressed as percentage of total spermatozoa analyzed, are presented as mean ± SEM. ** *p* < 0.01.

**Table 2 ijms-25-01682-t002:** (**a**) The abundance relation of each metabolite in spermatozoa (CELL) and seminal plasma (SP) from NORMO group. ↑ increase. * *p* < 0.05 and ** *p* < 0.01; (**b**) overrepresented metabolic pathways using the phosphorylated metabolites assigned by ^31^P-NMR in SP and CELL from NORMO group. All the phosphometabolites identified through the correspondent human metabolome database (HMDB) codes and converted to Kyoto Encyclopedia of Genes and Genomes (KEGG) codes, were analyzed using the bioinformatics tool Metabolites Biological Role (MBROLE) to perform an overrepresentation analysis for the cellular pathways, based on the KEGG pathways database (https://www.genome.jp/kegg/pathway.html, accessed on 19 November 2022). Significant when *p* < 0.05. Note: GPL = glycerophospholipid.

(a)					(b)		
NMR	Metabolites	↑ CELL	↑ SP		Metabolic Pathway	*p* Value	Metabolites
**^1^H**	Citrate	1.28 **			**CELL**		
	CHO		1.30 **		Signal transduction	1.32 × 10^−5^	NMP|NMPc
	Tyr		1.25		Purine metabolism	3.34 × 10^−4^	NMP|NMPc
	LAC	1.07			Ether lipid metabolism	3.42 × 10^−3^	GP(N-biotin)E|GP(monomethyl)E
	Gln	1.57 **			Metabolic pathways	4.39 × 10^−3^	NMP|G1P|PCh|AcP|X5P FBP|S7P|GP(monomethyl)E GP(N-biotin)E
	Phe		1.01				
**^31^P**	GPC		1.11				
	NMP	2.12 *			PPP	6.57 × 10^−3^	X5P|S7P
	S7P and FBP	4.95			GPL metabolism	1.33 × 10^−2^	PCh|GP(monomethyl)E GP(N-biotin)E
	GPE	1.33					
	G1P	2.84 **			Pentose and glucuronate interconversions	1.74 × 10^−2^	G1P|X5P
	GPI		3.35 **				
	G6P		1.38		**SP**		
	PCh	3.01 **			GPL metabolism	6.67 × 10^−5^	GPC|GPI|GP(dimethyl)E
	PDE4	2.31 **			Ether lipid metabolism	8.11 × 10^−4^	GP(dimethyl)E
	PDE3	12.54 **			Inositol phosphate metabolism	2.34 × 10^−3^	G6P|GPI
	PDE2	1.17					
	GP(dimethyl)E		6.86 **		GPI-anchor biosynthesis	5.85 × 10^−3^	GPI
	GP(N-biotin)E	1.33			Regulation of autophagy	9.73 × 10^−3^	GPI
	X5P	3.14 **					
	PDE1	1.37					
	6PG		1.32				
	NMPc	1.26					
	AcP	1.35					
	NADPH		1.50				
	GP(monomethyl)E	1.06					

## Data Availability

The datasets generated and analyzed during the current study are available from the corresponding author upon reasonable request.

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
