# Peer review of "Quantitative Analysis of the Human Semen Phosphorometabolome by 31P-NMR"

_ijms, 2024, doi:10.3390/ijms25031682_

Round 1
Reviewer 1 Report
Comments and Suggestions for Authors
Title
I suggest change title to "Quantitative analysis of the human semen phosphorometabolome by 31P-NMR."
Abstract:
You are not providing any data on sperm quality measures
Keywords
Please do not use keywords that have already been mentioned in the title. This increases the possibility of searching your manuscript.
Introduction
It is appropriate, but since you have looked into a number of sperm characteristics, it would also be helpful to have a brief introduction to the other parameters in addition to motility.
Conclusions
The conclusion should be more specific to your findings
Author Response
Title
I suggest change title to "Quantitative analysis of the human semen phosphorometabolome by 31P-NMR."
Response:
The title has been changed as suggested
Abstract:
You are not providing any data on sperm quality measures.
Response:
We have included the values obtained in total and progressive sperm motility in both sperm populations (Abstract lines 4-5). We did not include them in the first version of the manuscript due to the Abstract words restriction. So, the normozoospermic samples that we used in this work presented an average of 71% of total motility and 45% of progressive motility, whereas asthenozoospermic samples had 27% of total motility and 17% of progressive motility, according to the WHO.
Keywords
Please do not use keywords that have already been mentioned in the title. This increases the possibility of searching your manuscript.
Response:
As we have modified the title, now there are not keywords already mentioned in the title. We thank the referee for this useful comment.
Introduction
It is appropriate, but since you have looked into a number of sperm characteristics, it would also be helpful to have a brief introduction to the other parameters in addition to motility.
Response:
As suggested, we have included in the introduction (last paragraph) some sentences about other sperm parameters studied in both sperm populations, normozoospermic and asthenozoospermic: sperm viability, mitochondrial membrane potential and mitochondrial superoxide production.
Conclusions
The conclusion should be more specific to your findings.
Response:
We have rewritten the Conclusion, as requested, aiming to be more accurate to the data obtained.
Reviewer 2 Report
Comments and Suggestions for Authors
1. What new information regarding sperm functions is generated in this work? How it contribute to understand male infertility? The data do not provide clear cut to understand better metabolic mechanisms.
2.How metabolic components in the seminal plasma provides information to better understand metabolic mechanisms in sperm? It is written in lines 265-7 that the presence of sperm does influence the picture of metabolites in the semen?
Author Response
1. What new information regarding sperm functions is generated in this work? How it contributes to understand male infertility? The data do not provide clear cut to understand better metabolic mechanisms.
Response:
This work reveals new molecular information about the metabolic pathways underlying human sperm motility, by deciphering the phosphometabolome of human semen using 31P-NMR from donors with asthenozoospermia (low sperm motility, which according to WHO means < 40 % of total motility and/or <32 % of progressive motility) or normozoospermia. To date, the phosphorus-containing metabolites associated to high or low human sperm motility have not been investigated yet. Specifically, our data indicate that the phosphorometabolites produced by the metabolism of carbohydrates in human spermatozoa are more abundant that those produced by lipid metabolism. In fact, our results highlight specific metabolic alterations in spermatozoa with lower motility (asthenozoospermic samples), where an increase in the phosphometabolites acetyl phosphate, phosphocholine and glucose-1-phosphate is found. This finding suggests potential impairments in choline, taurine and glycerophospholipid metabolism occurring in asthenoszozospermic men and would allow focusing in a near future on the research of metabolic pathways involving mentioned phosphometabolites to discover which enzyme or reaction is malfunctioning.
In addition, this work contributes to understand human male infertility as sperm motility is considered by WHO a functional parameter indicator of seminal quality. We believe that unraveling the human sperm phosphorometabolome associated to motility would undoubtedly deepen the sperm metabolic pathways that are specifically impaired in asthenoszoospermia and therefore can be a cause of male infertility. Moreover, this study identifies potential molecular biomarkers to assess human sperm health and provides insights into the specific metabolic pathways altered in asthenozoospermia, one of the known causes of male infertility.
In our opinion, this work also lays solid groundwork for developing more precise diagnostic approaches of human fertility and contributing to a more personalized therapies in the future. Consequently, this work contributes to enhancing the molecular and metabolic understanding of male infertility and can, in a future, help to improve its clinical management
2. How metabolic components in the seminal plasma provides information to better understand metabolic mechanisms in sperm? It is written in lines 265-7 that the presence of sperm does influence the picture of metabolites in the semen?
Response:
In the manuscript we meant that the presence of spermatozoa in whole human semen does not significantly influence the phosphorometabolites profile of the ejaculate when compared with the obtained from seminal plasma (lines 265-267 of former manuscript). We have found that the phosphometabolic profile observed in ejaculates is similar to that of seminal plasma, indicating that the metabolism of spermatozoa, cells that represent less than 10% of total ejaculate volume, does not significantly contribute to the phosphorometabolic profile of the ejaculate.
We have revised the sentences mentioned by the reviewer in order to clarify this issue (formerly lines 265-267, now in lines 271-276 of this revised version). New sentence states as follows:
“Analogous results to seminal plasma were obtained when comparing whole ejaculates, which is reasonable because spermatozoa only represent <10% of the total human semen volume [43], being seminal plasma the major component of the ejaculate. Thus, we conclude that the metabolism of spermatozoa in whole human semen does not significantly contribute to the phosphometabolic profile that is observed in seminal plasma”
Round 2
Reviewer 2 Report
Comments and Suggestions for Authors
None
Author Response
In the web page web in “ Manuscript Information Overview" and then in "Review Report" we can see that referee 2 does not state any comment.